# Physical Condition Factors that Predict a Better Quality of Life in Women with Fibromyalgia

**DOI:** 10.3390/ijerph16173173

**Published:** 2019-08-30

**Authors:** Núria Sempere-Rubio, Marta Aguilar-Rodríguez, Marta Inglés, Ruth Izquierdo-Alventosa, Pilar Serra-Añó

**Affiliations:** 1Unidad de Biomecánica Clínica (UBIC Research Group), Department of Physiotherapy, Faculty of Physiotherapy, Universitat de València, 46010 València, Spain; 2Freshage Research Group, Department of Physiotherapy, Faculty of Physiotherapy, Universitat de València, Centro de Investigación Biomédica en Red Fragilidad y Envejecimiento Saludable (CIBERFES-ISCIII), Fundación Investigación del Hospital Clínico Universitario de Valencia (INCLIVA), 46010 València, Spain

**Keywords:** quality of life, fibromyalgia, functional capacity, cardiorespiratory fitness, muscle strength, postural balance, pain threshold, anxiety

## Abstract

What physical qualities can predict the quality of life (QoL) in women with fibromyalgia (FM)? QoL is a very complex outcome affected by multiple comorbidities in people with fibromyalgia. This study aims to determine which physical qualities can predict the quality of life in women with FM. Also, a comparison between the physical qualities of women with FM and healthy counterparts was conducted. In total, 223 women participated in this cross-sectional study, 123 with FM, with ages ranging between 45 and 70 years. The study was conducted at several fibromyalgia associations and specialized medical units. QoL was measured as the main outcome. In addition, functional capacity, muscular strength, maintenance of thoracic posture, postural control, flexibility, pain threshold, and anxiety were measured. Prediction of the QoL was conducted with multiple linear regression analysis and comparison between groups, using the Mann–Whitney U test. There were significant differences between groups in all the variables measured (*p* < 0.01). The multiple linear regression model showed that factors influencing QoL in women with FM for all the variables measured were functional capacity, handgrip strength and bicep strength, maintenance of thoracic posture, pain threshold, and anxiety (*R*^2^ = 0.53, *p* < 0.05). To conclude, women with FM show a significantly lower QoL than their healthy counterparts, and the factors that predict their perceived QoL are functional capacity, muscular strength, postural maintenance, pain threshold, and anxiety.

## 1. Introduction

Fibromyalgia (FM) is a common chronic pain condition that has a significant impact on quality of life (QoL) [1], possibly due to its interference with physical ability, function, work, and social activities [2]. Previous studies revealed that the global physical condition (PC) in FM patients is impaired, since, compared to healthy counterparts, strength is reduced [3], postural control is altered [4,5], body posture is misaligned [6], or functional capacity is poor [1]. Furthermore, another important factor closely related to PC is flexibility, which is known to influence postural maintenance [7] and was independently associated with pain in this population [8], which was also associated with postural maintenance [6]. All these altered physical conditions could, in turn, possibly affect their perceived QoL.

The etiology and pathogenesis of FM are still unknown, but the syndrome is claimed to be multifactorial [9] and, therefore, physiotherapeutic programs must contemplate different approaches in order to improve well-being and QoL in FM patients.

Physical exercise was proposed as a suitable intervention for a variety of chronic pain populations, including FM, with the purpose of reducing pain severity, improving PC, and enhancing QoL. Nevertheless, based on a recent review [10], the evidence on the effects of physical exercise is of low quality because of the small sample sizes, short length of the intervention programs, or the short follow-ups used in the studies. This, together with the lack of adherence [11] and the fatigue experienced with physical effort in this population, suggests the need for a thorough study of the correlations between QoL and PC to focus the interventions mainly on the development of the physical variables most related to QoL, avoiding fatigue as much as possible.

Physiotherapeutic interventions in people with FM most usually focus on the improvement of PC [12,13,14] and some other variables related to PC, such as anxiety [15], commonly present in people with FM, and those related to the perceived pain experience [16]. However, as mentioned above, it becomes necessary to establish which of the basic physical abilities or psychomotor qualities are more related to QoL and are likely to be addressed by physiotherapy. In this way, more personalized physiotherapy treatments could be carried out, providing better care for patients with FM.

This study was aimed at identifying predictors of QoL related to PC and anxiety. Additionally, it strove to determine the differences in PC of people with FM compared to healthy counterparts.

## 2. Methods

### 2.1. Participants

This cross-sectional study used purposive modal instance sampling to select the participants. The sample size calculation was conducted to ensure 80% of power, setting the type I error at 5%. A medium-size effect (d = 0.5) and seven independent variables were predicted. With these requirements, 103 people were required.

The FM group (FMG) was composed of 123 women diagnosed with FM who were recruited from several fibromyalgia associations and specialized medical units over a year and a half. The inclusion criteria for the FMG were women between 45 and 70 years of age diagnosed according to the 2010 American College of Rheumatology criteria [17]. The control group (CG) was composed of 100 age-matched women without symptoms.

Exclusion criteria for both groups were as follows: history of fractures or surgery over the past six months, inflammatory rheumatic disease, neurological disorder, peripheral neuropathy, or suffering any acute and terminal illness.

#### 2.1.1. Assessment Procedures

Several features of the participants’ PC were included as the dependent variables: functional capacity, muscular strength, maintenance of thoracic posture, postural control, and flexibility. Additionally, their pain threshold, QoL, and anxiety were measured.

#### 2.1.2. Quality of Life Measures

QoL was assessed using the Spanish version of the Short Form-36 Health Survey (SF-36) [18] whose internal consistency and reliability was validated (Cronbach’s α >0.70 and Intraclass correlation coefficient (ICC) >0.90, respectively) [19], and it is commonly used to measure physical, social, role, and emotional functioning. The score ranges from 0–100. Scores above or below 50 indicate better or worse health status, respectively, than the mean of the reference [20].

#### 2.1.3. Functional Capacity

The two-minute walk test (2MWT) was used to measure the functional capacity. This test consists of determining the maximum distance (in meters (m)) that can be walked in 2 min. This test shows a good correlation with the six-minute walk test (6MWT), which was extensively used as a reliable measure of functional capacity in individuals with a variety of characteristics [21,22], including individuals with fibromyalgia [8]. However, this test is less fatiguing and better tolerated [23]. It was carried out following the instructions of the study conducted by Johnston and colleagues [24].

#### 2.1.4. Upper Limb Strength

Maximal isometric strength was assessed using two portable dynamometers, NedDFM/IBV (Instituto de Biomecánica de Valencia, Valencia, Spain), to assess the bicep strength, and NedVEP/IBV (Instituto de Biomecánica de Valencia, Valencia, Spain), to assess the handgrip strength. To conduct these two measures, the participants remained seated without any back support and with their feet on the floor. When bicep strength was assessed, the individuals had to attempt to flex their elbow with their palm upward as hard as they could against the evaluator’s equal resistance without moving their trunk. The dynamometer was placed on the distal portion of the arm. When handgrip strength was assessed, the participants were required to tightly grasp the dynamometer as hard as they could. Three repetitions of each measurement were performed consecutively with a 30-s rest between them, and the mean of the three repetitions was calculated. The order of the two strength measurements was counterbalanced.

#### 2.1.5. Maintenance of Thoracic Posture

To measure the participants’ ability to maintain the thoracic position for 5 min, the change in thoracic angle in a sitting position was calculated. For this purpose, the participants sat with a computer placed in front of them with the upper frame of the screen just below the height of the eyes and handled the mouse with the dominant hand. Adhesive markers were then positioned on the tragus, C7, and the spinous process of the seventh thoracic vertebra (T7).

To measure changes in thoracic posture, a photograph was taken each minute, using the same procedure as described in a previous study of our group [6]. The photographs were also analyzed with the software ImageJ (National Institutes of Health, Bethesda, Maryland, USA) [25], and we computed the difference between the last and the first photograph. A higher angle of change denotes a poorer the ability to maintain the posture.

#### 2.1.6. Postural Control

The postural control test was performed using the Wii Balance Board (WBB) (Nintendo, Kyoto, Japan) force platform [26]. Subjects were asked to place their feet hip-width apart, toes pointing forward, and arms relaxed at their sides in all the tests. A reference point was situated 2 m in front of the subject at eye level. All the subjects had to maintain the bipedal standing position with their eyes open during the test. The subjects performed two consecutive 60-s repetitions, and the mean was used for subsequent analyses. They rested for 30 s between repetitions, unless they needed extra time. The procedures were further explained in a previous study of our group [5].

Two variables derived from this test were measured: (i) ellipse: a measure of the area that the centre of pressure (COP) traverses, determined by taking the radius of the major and minor axes and then fitting an ellipse that would include 95% of the points [27]; (ii) sample entropy (SampEn), indicating the regularity of a time series (i.e., COP path) by calculating the probability of it having repeated itself. The calculation of SampEn was conducted following the description given by Randami et al. [28].

#### 2.1.7. Flexibility

V-sit and reach (VSR) was used to measure the global flexibility of the participants [29]. The procedure requires that the individual be placed sitting on the floor in a V-sit position with their feet 30 cm apart. At the midpoint of that distance, the evaluator places a measuring tape starting from 23 cm. Two repetitions of each measurement were performed consecutively with a 30-s rest between them, and the mean of the repetitions was calculated.

#### 2.1.8. Pain Threshold

The pain threshold of the trapezius was measured with a Wagner FPK 20 algometer (Wagner Instruments, Greenwich, CT, USA) with a contact area of 1 cm^2^ applied perpendicularly to the skin following the protocol of Slater and colleagues [30], locating the upper trapezius in the mid-point between the C7 spinous process and the acromion. Three repetitions of each measurement were performed consecutively with a 60-s rest between them, and the mean of the repetitions was calculated.

#### 2.1.9. Anxiety

Anxiety was evaluated with the Hamilton anxiety rating scale (HARS), translated and validated in Spanish with good internal consistency [31]. Scores for the entire scale (emotional distress) ranged from 0–56, with higher scores indicating greater distress.

### 2.2. Data Analysis

Data analysis was performed using SPSS (Statistical Package for Social Sciences, SPSS Inc., Chicago, IL, USA) software, version 22. The Mann–Whitney U test was used to compare the previously described dependent variables between groups (i.e., FMG and CG) since the normality was not satisfied (using Kolmogorov–Smirnov analysis). This test was also used to compare the age and body mass index (BMI) between groups. Furthermore, multiple linear regression analysis, with the backward method, was used to determine the influence on the QoL (measured with the SF-36) of the variables thought to be most influential. The following assumptions required for this analysis were checked: (i) independence of observations with Durbin–Watson, (ii) linear relationship between the dependent variable and the independent variables using Spearman correlation analysis, and (iii) multicollinearity with variance inflation factor (VIF). A probability value of *p* < 0.05 was considered statistically significant.

### 2.3. Ethical Approval

The project was approved by the Ethics Committee on Human Research of the University of Valencia (reference number: H1449048793044). All enrolled participants provided informed written consent prior to the study. The procedures were performed in accordance with the principles of the Declaration of Helsinki.

## 3. Results

### 3.1. Participants

A total of 123 women with FM were studied with a mean (SD) age of 54.40 (6.75) years. The CG was composed of 100 women with a mean (SD) age of 54.27 (6.08) years. Nevertheless, due to a technical failure in the postural control test, only 114 women with FM completed all the measurements. Therefore, the statistical power was lower in the two variables derived from the postural control (Table 1), although 80% of power was assured. There were no significant differences between groups in age (*p* > 0.05). The BMI was significantly higher in the FMG than in the CG (mean difference = 1.73 points; U = 7.31, Z = −2.41, *p* = 0.02), although both groups belonged to the “overweight” category.

### 3.2. Between-Group Comparisons

Table 1 shows that there were significant differences between groups in all the variables measured (*p* < 0.01). The QoL, functional capacity, isometric strength (both bicep strength and handgrip strength), SampEn, flexibility, and pain threshold were significantly lower, whilst the changes in maintenance of thoracic posture, the excursion of center of pressure (represented by the ellipse), and anxiety were significantly higher (*p* < 0.01).

### 3.3. Quality of Life Prediction

Table 2 shows the results of the multiple regression model to estimate the factors that affect QoL in women with FM. As noted, of all the variables measured, those able to predict QoL in women with FM were as follows: functional capacity, handgrip strength and bicep strength, maintenance of thoracic posture, pain threshold, and anxiety (*R* = 0.73, *R*^2^ = 0.53, *p* < 0.05). The assumption of independence of observations was satisfied (Durbin–Watson = 1.89), as was the multicollinearity analysis (VIF ranged from 1.04–1.93). The linear relationships between the dependent and the included independent variables were significant (*p* < 0.05).

## 4. Discussion

The current study shows that functional capacity, upper limb muscular strength, postural maintenance, pain threshold, and anxiety are important predictive factors of QoL in women with FM. To the best of our knowledge, this is the first study that selected, from a variety of variables that can be treated by the physiotherapist, those that can predict changes in the QoL of women with fibromyalgia, in order to establish the appropriate therapeutic guidelines.

All the variables studied showed a significantly different score between women with FM and their healthy counterparts, as disclosed in the results. Women with FM were observed to have a poorer functional capacity, lower isometric strength, a poorer ability to maintain thoracic posture and to maintain postural control, as well as a lower pain threshold, reduced flexibility, and increased self-reported anxiety. Of these, however, postural control and flexibility did not show a significant contribution to QoL, as demonstrated by the regression analysis.

With regard to functional capacity, our findings are in accordance with previous studies showing that there are differences between women with FM and healthy women [1,3,32,33,34]. According to our study, functional capacity rendered a lower value in women with FM. This datum could be due, on the one hand, to lack of physical fitness, as these patients often adopt a more sedentary lifestyle [35], which might include frequently resting in bed when suffering from symptoms. On the other hand, this could be due to lower cardiorespiratory fitness in this population. In this regard, more accurate methods, such as the graded exercise test, to determine maximal oxygen uptake (VO_2_ max,) which is the gold standard for cardiorespiratory fitness (thus indicating the maximal aerobic power), would have been desirable. However, due to the particular characteristics of this population (i.e., pain, fatigue), such exhausting measures are often difficult to implement. Thus, submaximal field exercise tests, such as the two-minute walking test, provide a feasible, safe, easy-to-administer, and inexpensive technique for the prediction of VO_2_ max [36], and may be considered as an indirect measure of maximal aerobic power or cardiorespiratory fitness in this population. Indeed, the two-minute walking test was shown to have a moderate-to-strong correlation with VO_2_ max consumption [37]. Our results are in agreement with previous studies showing that the cardiorespiratory fitness of women with FM was even lower than that of healthy sedentary women [38]. In addition, previous studies observed that these people reflect lower respiratory muscle resistance, lower strength in inspiratory muscles, and less chest mobility, which in turn could contribute to lower aerobic capacity [39].

Our results suggest that an increase in functional capacity implies an increase in the QoL score in this population. There is some controversy regarding the relationship between functional capacity and QoL. Two previous studies did not show a clear relationship between QoL and functional capacity, although their sample was small [33,34]. However, Carbonell-Baeza et al., with a sample size similar to ours, found a relationship between functional capacity and QoL in FM [40]. Nevertheless, an adequate functional capacity is necessary to perform many daily life activities which imply a moderate level of physical activity. Diminished capacity could inevitably have an impact on the level of participation in this type of activity, which in turn would lead to a poorer QoL and even a dependence on other people [41].

With regard to muscle strength, previous studies found reduced muscle strength in women with FM compared to healthy women, both in grip strength [3,32,42], linked to a state of sarcopenia [43], and in upper limb strength [3,32], linked to functional limb capacity [44]. The results of our study are consistent with those of the aforementioned studies, since, in our study, muscle strength was lower in people with FM; specifically, isometric grip strength was 47% lower than that of healthy subjects, and isometric strength of the upper limb was 45% lower than the control group. This decrease in muscle strength could be due, as noted by previous studies, to physiological and neuromuscular factors typically found in FM, such as alterations in blood circulation and changes in neuromuscular control mechanisms caused by pain [45].

Our study shows that an increase in isometric muscle strength, both in grip and the upper limb, can predict an improved QoL. In fact, a given study already established in this population a direct relationship between grip strength and upper limb strength with QoL in FM patients [46]. This result implies that it would be useful for therapeutic intervention plans in this population to include strength training programs that could contribute to improve QoL. Furthermore, since FM women are over the age of 50, the strength training may contribute, in turn, to preventing or postponing as much as possible the onset of sarcopenia and eventual frailty [47].

Parallel to the loss of strength, there is evidence in our study of an inability of the postural muscles to maintain posture, as reflected in the variable maintenance of thoracic posture. Our group previously confirmed a relationship between pain and poor sitting posture in FM [6], as already established in populations with pain [48]. Postural disorders are observed in women with FM, such as positioning the head abnormally forward; these are aggravated by the muscular and joint rigidity of the spine present in this syndrome, thus increasing pain and preventing normal activities of daily and working life [49]. This difficulty in maintaining the upright posture of the trunk and neck may be due to altered muscle control strategies. This is because, by positioning the head abnormally forward, resistance of the muscles responsible for the upright posture deteriorates and peripheral muscle pain appears, inducing changes in the adaptation of the central nervous system and producing impaired control strategies [6]. Furthermore, some emotional states often present in women with FM, such as depression and anxiety, necessarily influence QoL [42] and may induce postural dealignment [50,51]. Indeed, depression significantly affects posture [52,53], as evidenced by an increased flexion thoracic kyphosis found in individuals with depression [53].

Our results show that a reduced ability to maintain trunk posture can predict a decline in QoL. The distribution of tonic muscle activity (“posture”] depends on the system of posture control [54]. In turn, postural control impairment can affect balance and the performance of daily living activities, and may eventually lead to falls [55], thus negatively affecting their quality of life, as reported in other populations [56,57]. In addition, stress and anxiety may affect postural control [5,58]. These findings are interesting for a better physiotherapeutic approach to patients with FM; neuromuscular control management through training with trunk and neck posture control exercises, as demonstrated in other pain populations, is an important point [48,59]. Furthermore, improving thoracic kyphosis and adopting an upright position was shown to reduce fatigue and stress [60,61], both key factors that should be improved in women with FM [2,62].

Other physical conditions that showed differences between women with FM and healthy subjects are balance and flexibility. However, none of these showed any significant influence on QoL. Our results, regarding the postural control and flexibility variables, are consistent with Kibar’s research [63], which conducted balance and flexibility training and found no improvement in QoL in women with FM. Previous studies suggested that balance in people with FM may be mediated by other factors such as anxiety, depression, and fatigue, and may, therefore, mask the relationship with QoL [55]. Future studies should analyze in greater depth the effect of balance alterations on QoL in this population to better understand the effect of possible confounding variables.

Parallel to the alteration of these physical properties, women with FM suffer from generalized pain, which affects the day-to-day activities of women with FM and their participation in society. Pain adversely affects their QoL and limits their daily life activities [64]. According to our results, a lower pain threshold implies a lower QoL score. These data are consistent with previous studies in FM [16,65] and chronic pain populations [65]. Previous studies showed that pain in FM patients can be reduced through physiotherapy intervention-based exercise programs with strength training or moderate aerobic exercise combined with health education, resulting in an improvement in QoL [15,66]. Thus, being aware of its impact on QoL, it is essential to address the subject not only using these approaches but also seeking for other specific intervention plans aimed at reducing pain, in order to improve as much as possible the quality of life of women with FM.

As discussed, chronic pain experience is linked to anxiety in these people [67,68]. Our study confirms that women with FM have significantly greater anxiety levels than their healthy counterparts. It was also noted from the regression analysis that anxiety predicts QoL, since an increase in anxiety results in a decrease in QoL. Although there are no previous studies that attempted to predict QoL in women with FM using these types of variables, there were approximations that measured the correlation between the two variables. In this respect, the study by Ozcetin failed to show a correlation between the global score in the SF-36 with anxiety; however, they did find a negative relationship between anxiety and scores related to the physical and somatic function subscales of the SF-36 QoL questionnaire [68], demonstrating the link between physical state and the emotional component. It is useful for the physiotherapist to know how to objectively quantify the anxiety reported by patients with FM, in order to improve their symptoms through regular physical training. Moderate aerobic training, as well as strength training in FM patients, was shown to be helpful in treatments to improve and reduce anxiety symptoms [15].

## 5. Limitations

Studies of this type involve some limitations worth considering. Firstly, this was a cross-sectional study and, although regression analyses were used to predict causal direction, the results should be taken cautiously. Nevertheless, it would still be useful to establish if differences in QoL between people with FM and the control group could to some extent be related to the variables described. Secondly, this was a modal instance sample of patients, in an attempt to get a representative or typical expression of fibromyalgia phenomenology in only one region of the world. Broader, more representative samples of patients could have advantages. Finally, we controlled only for physical confounding variables; therefore, other confounding variables could be involved.

## 6. Conclusions

The results obtained from this study show that women with FM present differences in their overall physical condition compared to their healthy counterparts. The predictive factors of QoL in women with FM are functional capacity, muscle strength, postural maintenance, pain threshold, and anxiety.

## Figures and Tables

**Table 1 ijerph-16-03173-t001:** Results of the comparison between women with fibromyalgia and their healthy counterparts.

	FMG		CG		Group Comparison	
	Mean (SD)	Median	Mean (SD)	Median	Z	*p*
QoL	32.49 (15.10)	30.97	78.76 (12.81)	83.33	−12.29	<0.01
Functional capacity (m)	162.06 (27.8)	165.00	212.36 (25.54)	210.00	−10.72	<0.01
Bicep strength (kp)	52.49 (18.68)	47.33	95.92 (25.67)	95.94	−11.35	<0.01
Handgrip strength (kp)	82.13 (56.86)	71.83	155.24 (49.8)	156.80	−8.40	<0.01
Maintenance of thoracic posture (cm)	4.83 (4.66)	4.18	3.13 (3.03)	2.55	−2.92	<0.01
Ellipse (mm^2^)	360.14 (495.2)	151.88	155.26 (171.88)	96.37	−4.20	<0.01
SampEn	0.69 (0.15)	0.69	0.77 (0.1)	0.78	−3.86	<0.01
Flexibility (cm)	−1.28 (12.65)	−2.00	9.82 (9.9)	11.00	−7.04	<0.01
Pain threshold (kg·cm^−2^)	2.61 (0.65)	2.40	4.16 (1.2)	4.18	−10.18	<0.01
Anxiety	28.88 (9.52)	29.00	9.96 (6.82)	8.50	−11.72	<0.01

FMG—fibromyalgia group; CG—control group; QoL—quality of life; SampEn—sample entropy; Z—Mann whitney u test z value; *p*—significance value.

**Table 2 ijerph-16-03173-t002:** Multiple regression model for quality of life in women with fibromyalgia.

Predictor for Quality of Life	Unstandardized Coefficient Beta	Standard Error	Standardized Coefficient Beta	Confidence Interval
Constant	22.53	8.85	-	4.99 to 40.07
Functional capacity	0.14	0.04	0.26	0.07 to 0.21
Bicep strength	0.06	1.72	0.07	−0.09 to 0.21
Handgrip strength	0.03	0.07	0.11	−0.02 to 0.08
Maintenance of thoracic posture	−0.22	0.02	−0.07	−0.65 to 0.22
Pain threshold	1.64	0.22	0.07	−1.77 to 5.05
Anxiety	−0.74	0.12	−0.48	−0.97 to −0.51

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
