# Peer review of "Physical Condition Factors that Predict a Better Quality of Life in Women with Fibromyalgia"

_ijerph, 2019, doi:10.3390/ijerph16173173_

Round 1

Author Response

Response to the reviewer 1

Manuscript review

QOL and predictive factors in women with fibromyalgia

Thank you for the opportunity to review this manuscript. The topic is of interest to clinicians and pain specialists and addresses an important component of fibromyalgia – quality of life. The authors have recruited a larger sample than is often reported for this area and should be congratulated for that. Overall the manuscript is well organised and well-written. The main concern is that there is a lack of justification for the role that the independent factors chosen to be considered in this study have in influencing quality of life. There are some areas that need clarification and a couple of moderate to minor concerns listed below. 

Title:

This needs to be reworked for clarification: do the authors mean “Physical condition factors that predict a better quality of life in women with fibromyalgia?” Could the authors please consider this and adjust for clarity.

Author response (AR): We thank the reviewer for his/her suggestion. Accordingly, we have changed the title to: “Physical condition factors that predict a better quality of life in women with fibromyalgia” for clarity.

Introduction

The introduction needs to provide the rationale underpinning the selection of independent variables such as postural maintenance, flexibility, and quality of life as measured by SF 36. Could the authors please consider the literature and incorporate the rationale for the choices they have made on independent variables?

AR: We totally agree with the reviewer’s opinion that the introduction should provide the rationale of including the selected variables. Such information has been therefore included in the introduction section (see lines 39-43).

Methods and results combined

Could the authors please state the validity and reliability of their QOL measure and what constitutes a low score?

AR: The required information (validity, reliability and threshold score) by the reviewer has been included. Such information has been therefore included in the Methods section (see lines 87-92): QoL was assessed using the Spanish version of the SF-36 (18) whose internal consistency and reliability is appropriated (Chronbach alpha > 0.70 and ICC > 0.90, respectively)(19) and it is commonly used to measure physical, social, role, and emotional functioning. The score ranges from 0 to 100. Scores above or below 50 indicate better or worse health status, respectively than the mean of the reference.(20).

Could the authors please review their understanding of the 2MWT. It would seem to be a measure of functional capacity, but unreliable as a measure of aerobic capacity? Could the authors please justify their use of it as a test of aerobic capacity or endurance?

AR: We have changed the term aerobic and endurance capacity to functional capacity throughout the manuscript for clarity. However, we consider that the reviewer deserves an explanation for this specific query. Indeed, the 6-minute walking test been extensively used as a reliable measure of functional capacity in individuals with a wide variety of characteristics, including individuals with fibromyalgia.(1) The 2-minute walking test shows a good correlation with the 6 – minute walk test (6MWT), being less fatiguing and better tolerated,(2,3) and reliability is suitable other populations, such as COPD patients.(4) Due to the particular characteristics of this population (i.e. pain, fatigue), more accurate methods, such as the graded exercise test, to determine maximal oxygen uptake (VO2 max,) which is the gold standard for cardiorespiratory fitness (thus indicating the maximal aerobic power), are often difficult to implement. In this regard, submaximal field exercise tests, such as 2-minute walking test, provide a feasible, safe, easy-to-administer, and inexpensive technique for the prediction of VO2 max,(5) and may be considered as an indirect measure of maximal aerobic power or cardiorespiratory fitness in this population. Indeed, 2-minute walking test has been shown to have a moderate-to-strong correlation with VO2 max consumption.(6). We have included a more detailed information on this issue in the discussion section (see lines 218-229).

It is not clear to this reader how maintenance of posture or postural control are factors that might influence quality of life. The link seems tenuous. Could the authors please provide a deeper justification of these measures in the methods section?

AR: According to the reviewer’s request, we have provided a deeper justification of this issue. However, we considered that this information would fit better in the discussion section. (See lines from 274).

Ethics approval is mentioned both at the start of the methods section and under the heading Ethics Approval at the end, could the authors please remove one so there is no repetition?

AR: We thank the reviewer for detecting the mistake. We have removed the ethics approval statement placed at the start of the Methods.

Could the authors comment on whether they corrected the statistics for multiple comparisons or not? AR: As there were only two groups, we did not adjust p-value for multiple comparisons.

Clarity and ease of reading would be enhanced by maintaining the same order when reporting results as when describing the methods. Could the authors consider this point and reorder their text under between groups comparisons to do this (flexibility and sampan)?

AR: We thank the reviewer for detecting the mistake. The text has been reordered. Now, we follow the same order used in Methods and in the tables (Results).

When reporting Mann Whitney U tests the authors need to report the degrees of freedom, the U value and p-value in a manner: U(df)=Uvalue, Z=zvalue, p-value.

AR: We have added the U value and reported the results as required by the reviewer.

One of the assumptions made during multiple regression analysis in SPSS is that you have independence of observations which can be checked by using the Durbin-Watson statistic. Could the authors please report the results of this or that they completed this check? For multiple regression another assumption is that there needs to be a linear relationship between the dependent variable (QOL) and each of the independent variables. Could the authors please report that this was checked by visual inspection of scatterplots or another method? Did the authors check for multicollinearity of the independent variables?

AR: All these assumptions were checked and satisfied. We have included the tests used in the ‘Data Analysis’ section and we have reported the results of these tests in the ‘Quality of life prediction’ section.

The authors did not report whether they had any outliers in the data. Could they please report what would be classified as an outlier and whether any we accounted for?

AR: We looked for outliers previously to the analyses. We detected 6 outliers (Z-score above 3) and they were excluded from the analyses.

Were the data normally distributed? How did the authors check this?

AR: The normality was checked using Kolmogorov-Smirnov test. Since it was not satisfied, we used non-parametric analyses (Mann Whitney U- test) to check the differences between groups. We have included information about the testing of the normality in the text.

Could the authors please explain why they have tabulated the unstandardized coefficient beta score. It is usually understood that these scores cannot be compared against one another where the independent variables are measuring on different scales such as in this study. Could the authors please justify their inclusion in the table? Could the authors please add the standard errors for the standardised coefficient beta?

AR: The unstandardized coefficients beta were included to ensure the replication. The unstandardized coefficient represents the amount of change in a dependent variable Y (QoL) due to a change of 1 unit of each independent variable. With these coefficients, clinicians/readers could reproduce the prediction of QoL. Further, we have added the standard errors for the coefficients beta as required by the reviewer.

Discussion

The issue of whether aerobic capacity was tested with the 2MWT is important in how the results are discussed and this reader would encourage the authors to cite the study that has validated 2MWT as an aerobic test otherwise it would be prudent to report the results within validated parameters. This is especially so because there is already controversy about whether differences in aerobic capacity exist between women with FM and healthy controls. 

AR: We have addressed this issue through the text since we have modified the name of the variable and its interpretation. Information about this could be read in Methods and Discussion sections, as stated before.

The discussion could be broadened to consider current thinking in the pathophysiological mechanisms that underpin fibromyalgia, for example – difficulty maintaining an upright posture may also be linked to feeling burdened by the disease as exampled by Rosario et al (2014) J Bodyw Mov Ther 18 (4), 540, or Kendler (2017) Mol Psychiatry 22 (11) 1539, and stress may also play a role Doumas et al (2018) Exp Brain Res 236, 305

AR: The reviewer is right in his/her appreciation. Accordingly, we have inserted the following two paragraphs in the new version of the manuscript

Line 269: “Furthermore, some emotional states often present in women with FM, such as depression and anxiety, necessarily influence QoL(42) and may induce postural dealignment.(50,51) Indeed, depression significantly affects posture,(52,53) as evidenced by an increased flexion thoracic kyphosis found in individuals with depression.(53))”

Line 280: “Furthermore, improving thoracic kyphosis and adopting an upright position has been shown to reduce fatigue and stress,(60,61) both key factors that should be improved in women with FM.(2,62)”

While replication is great in science the discussion states that previous studies have shown many of the effects found here: relationship between QoL and muscle strength – isometric and grip strength, lower pain threshold, and anxiety. The maintenance of posture is a new contribution, but the rationale for considering it as an independent variable that would have an effect needs to be more clearly stated. 

AR: We believe that after addressing the above issues, the rationale of considering it as an independent variable has been provided. See the introduction section (lines 33-37).

Limitations

A final comment might be that sampling was a modal instance, in an attempt to get a representative or typical expression of fibromyalgia phenomenology. This in itself is quite ambitious because the disorder is not homogenous. It would be prudent to iterate that this was the mode of convenience sampling as the bias with instance sampling may be construed differently from a sample of convenience. 

AR: We totally agree with the reviewer’s appreciation. Accordingly, we have rewritten the paragraph, to specify that sampling was a modal instance.

 REFERENCES

Pankoff BA, Overend TJ, Lucy SD, White KP. Reliability of the six-minute walk test in people with fibromyalgia. Arthritis Care Res.  2000;13(5):291-5. Alfano LN, Lowes LP, Dvorchik I, Yin H, Maus EG, Flanigan KM, et al. The 2-min walk test is sufficient for evaluating walking abilities in sporadic inclusion body myositis. Neuromuscul Disord. 2014;24(3):222–226. Bohannon RW, Bubela D, Magasi S, McCreath H, Wang Y-C, Reuben D, et al. Comparison of walking performance over the first 2 minutes and the full 6 minutes of the Six-Minute Walk Test. BMC Research Notes [Internet]. 2014 ;7(1):269. Leung ASY, Chan KK, Sykes K, Chan KS. Reliability, Validity, and Responsiveness of a 2-Min Walk Test To Assess Exercise Capacity of COPD Patients. Chest [Internet]. 2006;130(1):119-25. American College of Sports Medicine. ACSM’s health-related physical fitness assessment manual. Philadelphia (PA): Lippincott Williams and Wilkins. 2013. Bernstein ML, Despars JA, Singh NP, Avalos K, Stansbury DW, Light RW. Reanalysis of the 12-Minute Walk in Patients With Chronic Obstructive Pulmonary Disease. Chest.

Reviewer 2 Report

INTRODUCTION
The introduction is clearly and well structured, allows the reader to situate himself in the context of the study.

METHODS
The inclusion and exclusion criteria were clearly defined.

The sample size calculation must be explained and calculated

RESULTS

The results are clear and concise with appropriate statistical analysis been performed appropriately and rigorously.

The multiple regression is well planned. However, the authors should rewrite the table 2 to a better understanding. For example, the authors must specify the parameter with the model with R and R2.

DISCUSSION

The discussion is adequate and considers the wide available body of literature on either about this topic.

CONCLUSIONS

The conclusions in abstract and final text should only present the main findings. 

REFERENCES

Please include the doi number of the references

Author Response

Response to reviewer 2

INTRODUCTION

The introduction is clearly and well structured, allows the reader to situate himself in the context of the study.

Author's response (AR): We thank the reviewer for his/her words and comments. We are very grateful.

METHODS

The inclusion and exclusion criteria were clearly defined.

The sample size calculation must be explained and calculated

AR: The sample size calculation has been explained.

RESULTS

The results are clear and concise with appropriate statistical analysis been performed appropriately and rigorously.

AR: Thank you very much.

The multiple regression is well planned. However, the authors should rewrite the table 2 to a better understanding. For example, the authors must specify the parameter with the model with R and R2.

AR: We have added the standard error for the coefficients beta. R and R2 is specified in the section where the table is placed.

DISCUSSION

The discussion is adequate and considers the wide available body of literature on either about this topic.

 AR: Thank you very much.

CONCLUSIONS

The conclusions in the abstract and final text should only present the main findings.

AR: The conclusions have been reworded following the requirements.

REFERENCES

Please include the doi number of the references

AR: We thank the reviewer for detecting the mistake. We have included the doi number.

Reviewer 3 Report

Dear Authors,

It was a real pleasure to read your manuscript entitled: Quality of life (QOL) and predictive factors in women with fibromyalgia.

I would like to congratulate you on preparing a manuscript addressing a very interesting topic. The most of issues are presented in an interesting way. I have not any suggestions and in my opinion the article is ready for publication without any corrections

Author Response

Dear Authors,

It was a real pleasure to read your manuscript entitled: Quality of life (QOL) and predictive factors in women with fibromyalgia.

I would like to congratulate you on preparing a manuscript addressing a very interesting topic. The most of issues are presented in an interesting way. I have not any suggestions and in my opinion the article is ready for publication without any corrections

Author's response: Thank you very much for your words.

Round 2

Reviewer 1 Report

Thank you to the authors who have done a thorough job of responding to earlier suggestions.

This reader would like to see one more english edit of the paper just to ensure clarity of message. Some of the new sections have minor grammatical errors and correcting would add to the legibility of the paper. 

The placement of periods (full stops) commas and numbered references needs to be tidied up - equal spaces and the determination of whether the numbers are before or after the punctuation mark. 

Both the introduction and discussion have been substantially improved.

I would recommend publication.